# Metagenomic Sequencing Revealed Differences in the Composition of Cecal Microbes in Different Breeds of Chickens

**DOI:** 10.3390/ani14010028

**Published:** 2023-12-20

**Authors:** Dan Yan, Guohui Li, Huiyong Zhang, Qian Xue, Chenghao Zhou, Yixiu Jiang, Jianmei Yin, Zhixiu Wang, Wenming Zhao, Wei Han

**Affiliations:** 1Jiangsu Institute of Poultry Science, National Chickens Genetic Resources, Yangzhou 225125, China; 15380368963@189.cn (D.Y.); sahui2008@163.com (G.L.);; 2College of Animal Science and Technology, Yangzhou University, Yangzhou 225000, China

**Keywords:** metagenomics, local chickens, gut microbes, breeds

## Abstract

**Simple Summary:**

In recent decades, the correlation between gut microbiota and the performance of animals has been widely examined. However, there is still a lack of research about the influence of different host breeds. We used five local Chinese chicken breeds as experiment materials to establish the relationship between the characteristics of hosts and the composition of microorganisms. In this study, metagenomic sequencing was used to explore the differences in the intestinal microbial composition of different breeds. Then, the functions of differentially expressed microorganisms were analyzed. Combined with previous research, we discussed the links between microorganisms and different breeds. Our study explains the relationship between the gut microbiome and the host breeds of local chickens, and reveals the specific flora in the gut of each breed of local chicken, which lays a foundation for the exploration of high-quality traits and the improvement of feeding conditions in the future.

**Abstract:**

Intestinal microorganisms are closely related to the health, development, and production performance of animals and can also provide basic information for us to fully understand and explore the germplasm characteristics and traits of local chicken breeds. In this experiment, we used five local Chinese chicken breeds as research materials; metagenomic sequencing technology was used to explore the differences in the composition of intestinal microorganisms in different breeds, and it was found that each local chicken breed had unique unigenes, of which Silkies had the most, with a total of 12,948. GO and KEGG analyses found that the biological processes most enriched by differential functional genes include genetic coding, macromolecular transport, protein synthesis, and molecular functions such as glycoprotein binding, protein hydrolysis, etc. Each breed is enriched with specific pathways, such as Anyi tile-like gray chickens, which are enriched with pathways related to disease resistance, while Gamecocks’ enrichment is related to amino acid metabolism. Random Forest and LEfSe analyses revealed specific species of intestinal microorganisms in the cecum of different breeds, such as *Exiguobacterium*, which is associated with melanin deposition in Silkies. Therefore, we infer that gut microorganisms are closely related to the formation of chicken breed characteristics, and the results of this experiment can provide a theoretical basis for the discovery of high-quality traits and the improvement of feeding conditions in the future.

## 1. Introduction

Microorganisms are widely distributed across various environments, animal surfaces, and intestines and are an indispensable part of daily life. The composition of the gut microbiome is influenced by a variety of factors, including species, the environment, sex, and diet [1]. Gut microbes also have an impact on the hosts’ physiology and immune function. For example, probiotics can improve feed digestion by producing phytase, lipase, amylase, and protease [2] or stimulate the gastrointestinal tract to secrete digestive enzymes [3]. Some strains increase the nutritional value of feed by producing vitamins, exopolysaccharides, and antioxidants [4]. Some gut microbes also produce a variety of bacteriocins with different functions, which can regulate the number of pathogens in the gastrointestinal tract and maintain the balance of the intestinal flora [5,6]. Nowadays, due to the overproduction and abuse of antibiotics, pathogens are becoming resistant to antibiotics much more quickly than new antimicrobial compounds can be discovered [7], and natural competition between microorganisms has become a new method of disease resistance. *Ligilactobacillus salivarius* was cultured and supplemented in the food of broiler chickens by Jiajun Yang to evaluate the effect of probiotics on growth through the modulation of gut microbial composition and its functional metabolites using metagenomic and metabolomic assays. They found that *Ligilactobacillus salivarius* optimized the microbial composition of the caecum of broilers [8]. Hailiang Yu used metagenomic and transcriptome sequencing techniques to identify microbiota and genes in the cecal contents and cecal tissue of infected (JS) and control (JC) chickens at day 4.5 post-infection, respectively. After E tenella infection, the abundance of *Lactobacillus*, *Roseburia* sp., and *Faecalibacterium* sp. significantly decreased (*p* < 0.05), while the abundance of *Alistipes* and *Prevotella pectinovora* increased significantly (*p* < 0.05) [9]. Rachel Gilroy conducted metagenomic sequencing of 50 chicken feces samples from two breeds, and cultured and genome-sequenced bacterial isolates from chicken feces, documenting over forty novel species, together with three species from the genus Escherichia [10].

Compared to mammals, poultry have shorter intestines. The time it takes for food to pass through the entire intestine of a chicken is less than 3.5 h, and the time it takes for food to reach each section of the intestine is different. This causes a discrepancy in the composition of microorganisms in different parts, which provides conditions for us to understand the correlation between digestion and microorganisms in poultry. Wen et al. analyzed the characteristics of microorganisms in the duodenum, jejunum, ileum, cecum, and feces of 206 yellow broilers under the same feeding conditions and found that *Methanobrevibacter* and *Mucispirillum schaedleri* were significantly correlated with fat deposition [2]. This provides strategies for altering the gut microbiota to control fat deposition. Zhao et al. examined 60 two-way-selected chicken families with a 10-fold difference in body weight at 56 days of age and performed high-throughput sequencing of their intestinal microorganisms. They detected 190 microbial strains, 68 of which differed significantly in different families and sexes [11].

In this study, we used the cecum contents of five local Chinese chicken breeds as research materials and performed metagenomic sequencing to investigate the differences in the intestinal microbial composition of different breeds of local chickens. Our results provide new ideas for the exploration and utilization of the characteristics of local Chinese chicken resources and the improvement of feeding conditions.

## 2. Materials and Methods

### 2.1. Animal and Sample Collection

Five different local chicken breeds (Silkies, Huainan Partridge chickens, Henan Gamecocks, Luyuan chickens, and Anyi tile-like gray chickens) were selected from the National Local Chicken Breed Gene Bank of National Chickens Genetic Resources (Yangzhou, China), comprising a total of thirty healthy hens, six for each breed. All the chickens in this experiment were hatched at the same time, and then we injected vaccines into them on the same day. After they grew up, all the hens were raised in individual cages. All chickens were allowed ad libitum access to water and food. All cages were placed in one breeding room. At 300 days of age, cecal tissue contents were taken, frozen with liquid nitrogen, and stored in a −80 °C refrigerator for later use. Ethical approval of animal survival was given by the Poultry Institute, Chinese Academy of Agricultural Sciences (Yangzhou, China) with the following reference number: CACJIPS01453.

### 2.2. Library Construction

In this study, we extracted DNA from cecal contents by using the E.Z.N.A.^®^ Stool DNA Kit (D4015-02, Omega, Inc., Norcross, GA, USA) according to the instructions provided, eluting total DNA in 50 μL of an elution buffer using the modified procedure (QIAGEN, Germantown, MD, USA) described by the manufacturer. After elution, DNA was stored at −80 °C until LC-BIO TECHNOLOGIES used PCR (LC-BIO, Hangzhou, China) for measurements.

DNA libraries were constructed using the TruSeq Nano DNA LT Library Prep Kit (FC-121-4001, Illumina, Hayward, CA, USA). Library construction starts with fragmented cDNA. Blunt-ended DNA fragments were generated using a combination of packing reactions and exonuclease activity, and size selection was performed using the sample purification beads provided. Then, A bases were added to the blunt end of each strand to prepare them for their connection to the index linker. Each adapter contains a T-base overhang, which is used to attach the adapter to the A-tail fragmented DNA. These adapters contain the full complement of sequencing primer hybridization sites for single, paired-end, and indexed reads. Single- or dual-index adapters are ligated to the fragments, and the ligated products are copied via PCR using the following process: initial denaturation at 95 °C for 3 min, 8 cycles of denaturation at 98 °C for 15 s, annealing at 60 °C for 15 s, extension at 72 °C for 30 s, and then a final extension at 72 °C for 5 min.

### 2.3. Sequencing and Biostatistical Analyses

NovaSeq 6000 was used for high-throughput sequencing in PE150 mode. The data were preprocessed, spliced, and assembled; then, the low-quality and host sequences were removed via fqtrim v0.94 using a sliding-window algorithm. All coding regions (CDS) of metagenomic contigs were predicted via MetaGeneMark v3.26. CDS sequences of all samples were clustered using CD-HIT v4.6.1 to obtain unigenes. Unigene abundance for a certain sample was estimated with TPM based on the number of aligned reads from bowtie2 v2.2.0. The non-redundant gene set was obtained using sequence clustering, and unigenes were obtained. The unigenes were compared with the NR_meta database to obtain species’ taxonomic information at all levels.

After annotating the information at various levels, we calculated the alpha diversity using QIIME 1.1, then used UniFrac distances to estimate beta diversity. The unigenes were compared with the GO, KEGG (v87.1), and other databases to obtain annotation information about each database. The abundances and differences were analyzed with regard to species classification, functional annotation, and genes, and the genes with differences were enriched using the GO and KEGG databases. Random Forest and LEfSe analyses were used to identify the key strains with different expression levels in the cecum of each species.

## 3. Results

### 3.1. Description of Metagenomic Sequencing Data and Gene Abundance

A total of 30 hen cecal tissue content samples were collected from five different local chicken breeds (Silkies, Huainan Partridge chickens, Henan Gamecocks, Luyuan chickens, and Anyi tile-like gray chickens). Metagenomic sequencing was performed using NovaSeq 6000 to obtain 1,233,468,818 raw reads. A total of 1,159,716,662 clean reads were obtained from 30 hens. After completing a genome assembly of sequencing data from each sample using IDBA-UD, a CDS prediction was performed on contigs with a length greater than 500bp. As for the distribution of unigene reads on each sample, the number of unigenes in Silkies far exceeds that of other breeds, as they displayed 12,948 unigenes that could be found via variety classification. There were 3673 unique unigenes in the Huainan Partridge chickens, 7244 in the Henan Gamecocks, 1689 in the Luyuan chickens, and 5645 in the Anyi tile-like gray chickens. The number of unigenes that exist across the five breeds was 749,319 (Figure 1).

### 3.2. Analysis of Main Flora

In this experiment, the sequencing genes were compared and annotated. It is clear that the majority of microorganisms were bacteria, but there were still viruses, archaea, eukaryotes, and some unclassified species. The results show that Firmicutes, Proteobacteria, and Bacteroidetes accounted for the largest proportion of microorganisms in the cecum of local chickens at the phylum level. At the genus level, the dominant bacteria genera were *Bacteroides*, *Escherichia*, *Fusobacterium*, *Faecalibacterium,* etc., and the predominant flora at the level of species were *Escherichia coli*, *Fusobacterium mortiferum*, *Bacteroides* sp. *An322*, *Enterobacteriaceae,* etc. (Figure 2).

### 3.3. Prediction of Gene Function

The GO database was used to predict the function of the unigenes. The biological processes that were predominantly enriched included genetic coding, macromolecular transport, protein synthesis, and molecular functions, such as glycoprotein binding, protein and amino acid binding, ribosome function, ATP binding, enzyme activity, DNA binding transcription factor activity, transmembrane transport, protein hydrolysis, etc. (Figure 3).

The KEGG database was used to predict the function of the sequenced genes. The enriched KEGG pathways mainly included carbohydrate metabolism, amino acid metabolism, cofactor and vitamin metabolism, translation, membrane transport, nucleotide metabolism, energy metabolism, replication and repair, signal transduction, glycan biosynthesis and metabolism, folding, classification and degradation, and the cell community: prokaryotes, lipid metabolism, the metabolisms of other amino acids, drug resistance, etc.

### 3.4. Alpha Diversity Analysis

Alpha diversity refers to the diversity within a specific environment or ecosystem, which is mainly used to reflect the richness and evenness of species, often reflected by Chao1, Observed species, Good’s coverage, Shannon, Simpson, and other indices. In the results of this study, the Chao1 and Observed species indices reflect that the species richness in the cecum of Silkies was the highest, while that of the Anyi tile-like gray chickens was the lowest. The Good’s coverage results show that the unigenes were abundant in all the samples. Simpson and Shannon indices show that the species richness of Silkies was the highest and that of Henan Gamecocks was the lowest (Figure 4). All results showed that the species richness of Silkies is the highest among five local chicken breeds, and the microbial coverage of each sample was very high.

### 3.5. Beta Diversity Analysis

Beta diversity reveals the species diversity between different environmental communities, usually via the initial calculation of the distance matrix between environmental samples, which comprises the distance between any two samples and is mainly determined with principal component analysis (PCA). PCA, principal coordinates analysis (PCoA), and other methods are used to observe the differences between samples. The results of PCA and PCoA showed that the Silkies had the most consistent population, followed by Huainan Partridge chickens. The other three local chicken populations all had significant individual differences (Figure 5).

### 3.6. Differential Flora Analysis

There are many methods to determine the different abundances of flora in each breed. In this study, we determine the differential flora in three ways. The first is the Kruskal–Wallis test, which is suitable for comparison. It is clearly evident that at the level of phylum, *Deferribacteres*, *Bacteria_unclassified*, *Candidatus_Berkelbacteria*, *Candidatus_Kapabacteria* and *Candidatus_Blackburnbacteria* show great variation among the five breeds. At the level of genus, the flora which are most abundant in Silkies are *Porphyromonadaceae* and *Mucispirillum*; in Huainan Partridge chickens, *Duodenibacillus*; in Luyuan chickens, *Shigella*, *Mucispirillum* and *Kagunavirus*. At the level of species, *Lachnoclostridium* sp. and *Mucispirillum schaedleri* show higher levels in Silkies. The Huainan Partridge chickens show a higher abundance of *Bacteroides unclassified*, *Bacteroides plebeius*, *Bacteroides* sp. *An269*, *Parabacteroides distasonis* and *Bacteroides plebeius CAG:211*. The flora that are most abundant in Anyi tile-like gray chickens are *Enterococcus unclassified* and *Firmicutes bacterium AM29−6AC* (Figure 6).

Random Forest (RF) analysis reveals key species that could distinguish the differences between the groups of samples. The results showed that among the top 20 strains that contributed significantly to the accuracy of the group, the ones enriched in the Anyi tile-like gray chickens were Streptococcus virus, *Alkalitalea saponilacus*, *Turicibacter* sp. HGF1, *Bacillus* sp., *Clostridium haemolyticum*, *Paracoccus*_sp._BP8, *Clostridium*, *Paenibacillus*_sp., *Exiguobacterium*, and *Vibrio* sp. from Huainan Partridge chickens. Those from the Henan Gamecocks included *Vibrio* sp. and *Exiguobacterium* sp.

LEfSe was used to analyze the strains with significant differences in abundance between the groups. The results showed that the contents of *Lactobacillus aviarius* and *Mucispirillum schaedleri* were significantly higher in the Silkies. *Bacteroides* were more abundant in the Huainan Partridge chickens. *Proteobacteria* was highly abundant in the cecum of the Henan Gamecocks. *Lactobacillus salivarius* was present in the Luyuan chickens. The contents of *Firmicutes_bacterium_AM29_6AC*, *Turicibacter*, *Aeromonas* and *Roseburia* were higher in the Anyi tile-like gray chickens (Figure 7).

### 3.7. Functional Analysis of Differentially Expressed Genes

Through the enrichment analysis of specifically expressed genes in different groups, it was found that the main enriched GO entries of these differentially expressed genes are predominantly related to protein synthesis and transmembrane transport, amino acid metabolism, glycosyl compound metabolism processes, and various digestive and respiratory enzyme activity. The KEGG pathway includes bacterial chemotaxis, a two-component system, ascorbate and aldarate metabolism, geraniol degradation, phosphotransferase system (PTS), xylene degradation, dioxin degradation, cationic antimicrobial peptide (CAMP) resistance, porphyrin and chlorophyll metabolism, glycosphingolipid biosynthesis (lacto and neolacto series), phenazine biosynthesis, vancomycin resistance, pentose and glucuronate interconversions, the biosynthesis of siderophore group’s non-ribosomal peptides and caprolactam degradation (Figure 8).

## 4. Discussion

Intestinal microorganisms are closely associated with the health, development and growth of animals. Understanding how breeds’ differences affect the composition of their intestinal microbiome will help improve animals’ health and production performance. Due to the particularity of chicken intestinal digestion, food stays in the cecum for a long time, creating a good habitat for cecal microorganisms. This experiment analyzed the differences in the composition of cecal microorganisms found in five local chicken breeds in China, and the authors determined the differentially expressed microorganisms and the functions of their genes and inferred a correlation between microorganisms and different local chicken breeds. Salanitro et al. analyzed the intestinal microbial metagenome of 7-, 14-, 21- and 42-day-old chickens and found that with age and growth, the pathogenic microorganism content of clostridium in the cecum increased significantly, while the number of probiotic lactic acid bacteria decreased sharply. The microbial diversity of the cecum of chickens is significantly greater than that of the ileum. The ACE and Shannon indices (diversity indices) increase with the age of chickens [6]. This shows that microbial richness and evenness increase with age. In the five local chicken breeds, we found the presence of pathogenic microorganisms, such as *Escherichia coli*, *Streptococcus*, *Vibrio* and *Clostridium*, which is consistent with Salanitro’s research. This may be due to the age of our chickens. Subsequently, we will examine the correlation between chickens’ growth performance and microorganisms, constructing an effective model to compare microorganisms and growth performance. This will be very significant in improving chicken feeding and management.

Local Chinese chicken breed resources are abundant. In the process of selective evolution, the formation of unique biological breed characteristics and the presence of local chicken intestinal microbial communities and hosts can be found, as in long-term co-evolution; therefore, the study of the characteristics of the intestinal microbial community of local chicken breeds is of great significance for us to fully understand the characteristics of the breed and promote the healthy breeding of poultry.

Silkies are a local chicken breed that originated in China, and they are also popular in many countries because of their peculiar appearance and docile character. They have silky feathers, black skin, black feet, and black intestines. In many Asian countries, people raise them for their delicious meat. But in some other areas, people also keep them as pets, and they can coexist well with other breeds. They are highly sought after in China for their value in traditional medicine [12]. Traditional Chinese medicine believes that pills made from Silkies can treat gynecological diseases. *Mucispirillum schaedleri* is related to the formation of the intestinal mucus layer [13]. Intestinal mucus plays an important role in protecting epithelial surfaces from pathogens, supporting commensal bacterial colonization, maintaining a proper digestive environment and promoting nutrient absorption [14]. *Myxospirillum* belongs to the phylum *Ferrobacteria* and is a common but low-abundance member of the microbiota in rodents, pigs and humans [15]. However, this strain can inhibit and interfere with the expression of salmonella invasion genes, and thus, cause enteritis [16]. The higher abundance of *Mucispirillum schaedleri* in Silkies may be one of the reasons for the distinctive coloration of the intestines of this breed. In 2023, Petra R. Quezada-Rodriguez confirmed that *Exiguobacterium* is strongly related to melanin deposition in Atlantic salmon gill tissue [17]. In this study, we found that the Random Forest analysis results showed that *Exiguobacterium* is highly enriched in the cecal intestines of Silkies. The guts of Silkies appear black, which may be related to the enriched microorganisms within them. Some consumers believe that the unique color means they provide special nutrition; with further study, we can try to establish the correlation.

In previous studies on hen intestinal stem cells, other scholars found that *Lactobacillus* has a regulatory effect on intestinal stem cells (ISCs), can drive the differentiation of ISCs, and directly promotes the activity of intestinal stem cells [18]. In a study on the late egg production performance of laying hens, it was discovered that *Lactobacillus* can reduce the abdominal fat deposition of birds and improve their egg quality, which is beneficial to the intestinal health and production performance of layers [19]. Luyuan chickens are known for their large and muscular body size and have good reproductive performance [20], which may be related to the enrichment of *Lactobacillus salivarius* in their bodies.

*Roseburia* is an anaerobic, Gram-positive probiotic that secretes butyrate that promotes anti-inflammation and regulates the metabolism, thereby improving the hosts’ health [21]. *Roseburia* also has the ability to ferment xylan and beta-mannan, which are common fiber components in the diet and important components of plant cell walls. In previous studies on the intestinal microorganisms of broiler chickens, it was found that Roseburia is more dominant in the intestines of newborn chicks than it is during the late embryonic stage, which possibly occurs to increase vitality in the early post-hatching stage [22,23]. *Turicibacter* is an anaerobic bacterium found in the intestines of many species that is capable of interacting with host-secreted bile acids and plays a role in neurotransmitter regulation in mammalian intestines [24]. In studies on the chicken digestive tract, it was found that *Turicibacter*’s response to whole bile and bile acids is complex, involving proteins from multiple pathways [25], and plays an important role in regulating the host’s energy metabolism. Anyi tile-like gray chickens adapt to roughage [26] and free-range farming. The advantages of these methods may be due to the high abundance of these two bacteria in their intestines, which improves the efficient utilization of roughage for Anyi tile-like gray chickens.

*Streptococcus virus* is a kind of phage that can specifically infect *Streptococcus*. *Streptococcus* is a Gram-positive coccus that inhabits the oral cavity of several animals, including humans. Most of them are not pathogenic, but some species of *Streptococcus* are associated with sepsis and respiratory infections [27]. Among poultry, *Streptococcus* isolated using Acsa Igizeneza from the Kariokor slaughterhouse has a pathogenicity of approximately 35% [28]. Due to the fact that free-range conditions are more unstable than cage-free conditions, Anyi tile-like gray chickens have good stress and disease resistance. This may be related to the high abundance of streptococcal phages in their guts.

The result of the KEGG analysis shows that the amino acid metabolism-related pathways of Henan Gamecocks appeared to be significant (*p* < 0.05). This characteristic may be related to the aggressive nature of this breed. After long-term artificial selection, Henan Gamecocks have developed different muscle distributions from other breeds [29]. Compared with broiler chickens, Henan Gamecocks have a stronger metabolism to support their activities of moving faster and fighting with each other. As a result, the composition of the intestinal microbial community also shows significant differences, such as higher abundances of *Proteobacteria* and *Vibrio*.

## 5. Conclusions

Our study used metagenomic sequencing to further explore the correlation between the host and the intestinal microbiota. Specific bacterial groups that are closely related to the formation of the characteristics of the breed were found in each local chicken breed, such as *Exiguobacterium* in Silkies and *Roseburia* and *Turicibacter* in Anyi tile-like gray chickens. Through alpha diversity, we found that the species richness of Silkies is the highest among five local chicken breeds. Our study provides a reference for the specific formation, future development, and utilization of resources.

## Figures and Tables

**Figure 1 animals-14-00028-f001:**
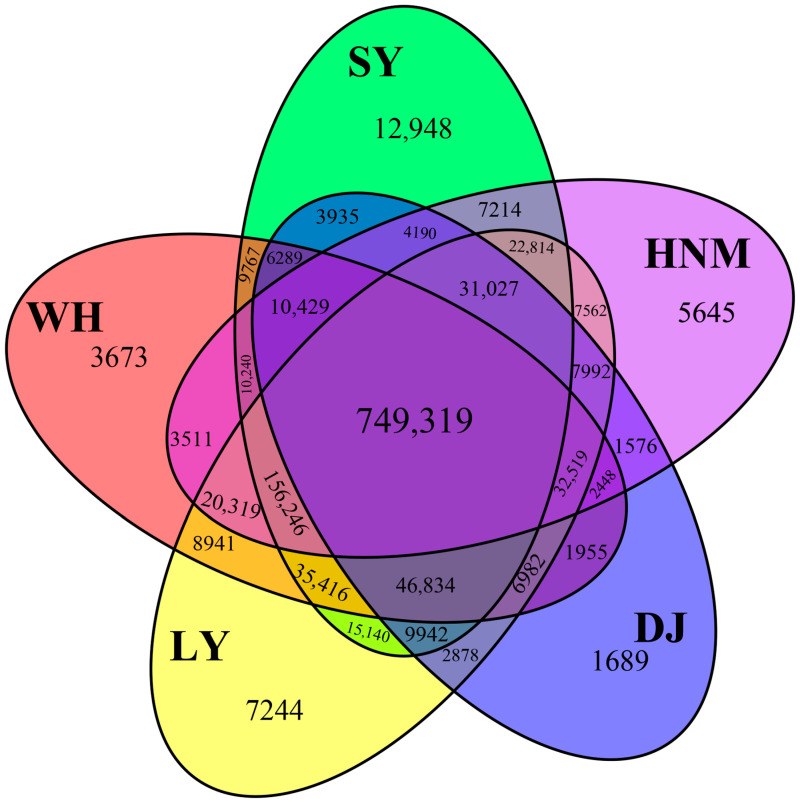
Venn diagram of unigenes enriched by different local chicken breeds. SY represents Silkies, HNM represents Huainan Partridge chickens, DJ represents Henan Gamecocks, LY represents Luyuan chickens, and WH represents Anyi tile-like gray chickens.

**Figure 2 animals-14-00028-f002:**
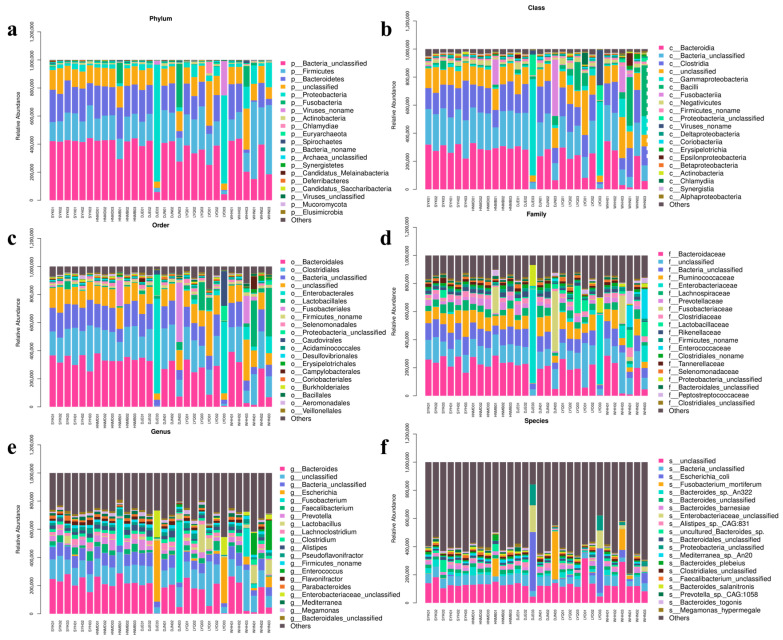
The proportion of cecal microorganisms enriched in different classification levels of chicken breeds in different places. In the picture, SY represents Silkies, HNM represents Huainan Partridge chickens, DJ represents Gamecocks, LY represents Luyuan chickens, and WH represents Anyi tile-like gray chickens. (**a**) Phyla level; (**b**) class level; (**c**) order level; (**d**) family level; (**e**) genus level; (**f**) species level.

**Figure 3 animals-14-00028-f003:**
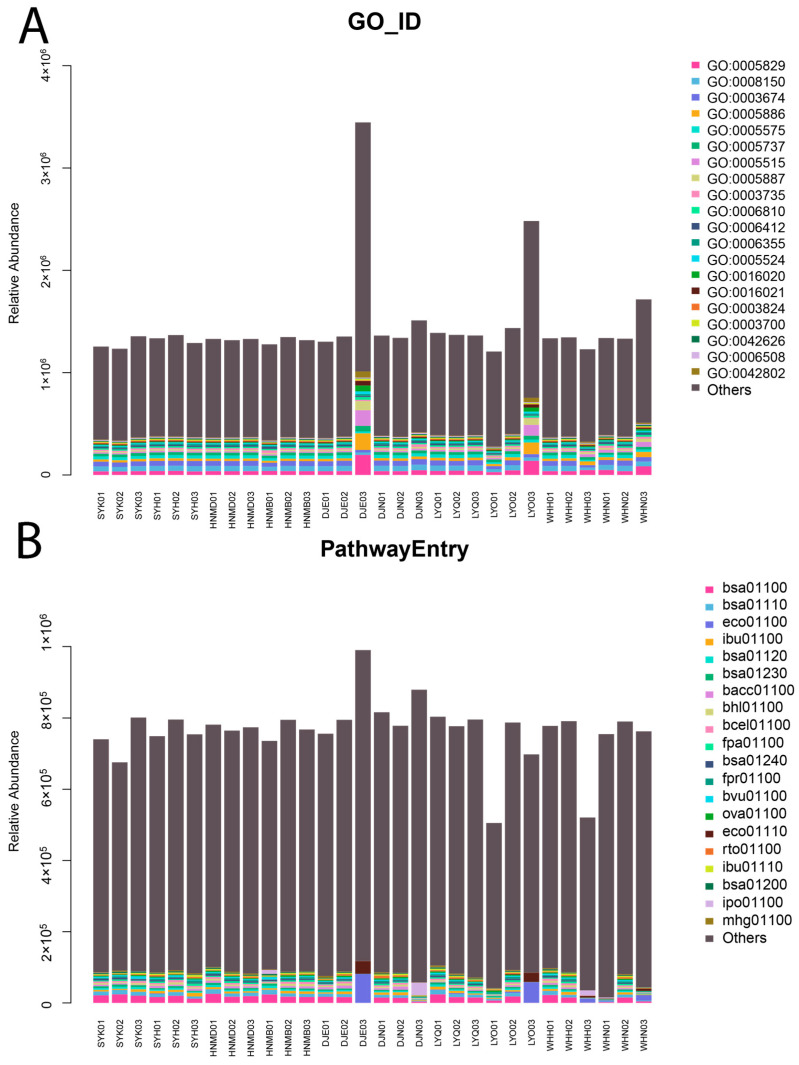
Column chart of gene function prediction for cecum microorganisms in each breed. (**A**): GO entry; (**B**): KEGG pathway. In the picture, SY represents Silkies, HNM represents Huainan Partridge chickens, DJ represents Gamecocks, LY represents Luyuan chickens, and WH represents Anyi tile-like gray chickens.

**Figure 4 animals-14-00028-f004:**
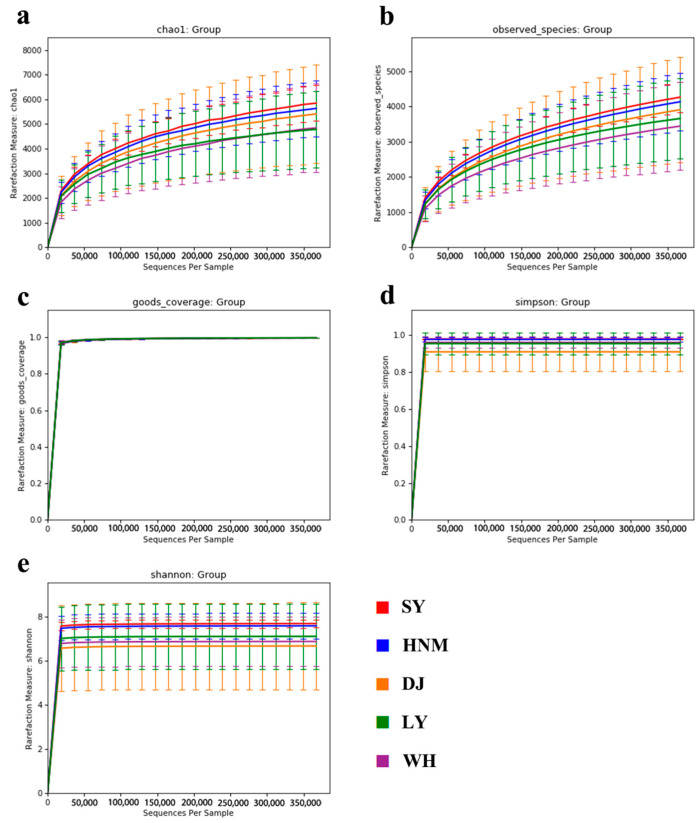
Dilution curve of microbial alpha diversity analysis in cecum of local chicken breeds. (**a**) Chao1; (**b**) Observed species; (**c**) Good’s coverage; (**d**) Simpson; (**e**) Shannon. SY represents Silkies, HNM represents Huainan Partridge chickens, DJ represents Henan Gamecocks, LY represents Luyuan chickens, and WH represents Anyi tile-like gray chickens.

**Figure 5 animals-14-00028-f005:**
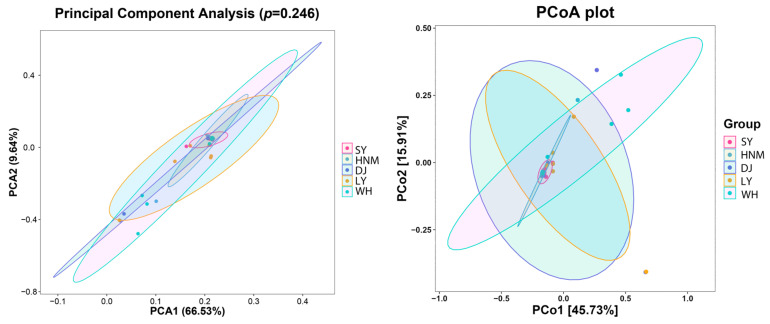
Beta diversity analysis of cecum microorganisms in different breeds. (**Left**): principal component analysis (PCA); (**right**): principal coordinate analysis (PCoA). A: Silkies; B: Huainan Partridge chickens; C: Gamecocks; D: Luyuan chickens; E: Anyi tile-like gray chickens.

**Figure 6 animals-14-00028-f006:**
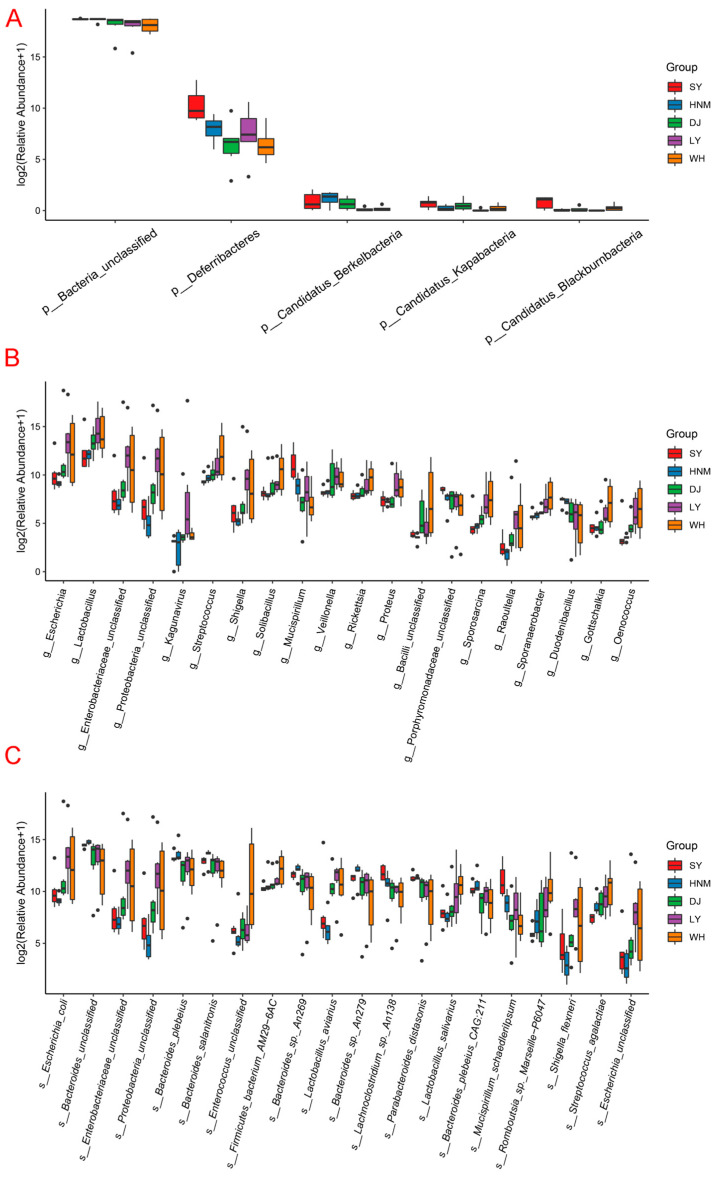
Flora with significant different abundance in the cecum of different breeds at each level. (**A**) Phylum; (**B**) genus; (**C**) species: SY represents Silkies, HNM represents Huainan Partridge chickens, DJ represents Henan Gamecocks, LY represents Luyuan chickens, and WH represents Anyi tile-like gray chickens.

**Figure 7 animals-14-00028-f007:**
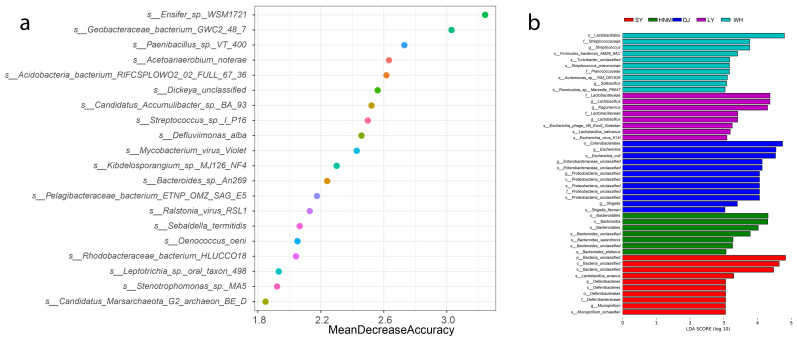
Flora with large abundance differences in the cecum of different breeds. (**a**) Random Forest analysis; (**b**) LEfSe analysis. In the right figure, A: Silkies, B: Huainan Partridge chickens, C: Henan Gamecocks, D: Luyuan chickens and E: Anyi tile-like gray chickens.

**Figure 8 animals-14-00028-f008:**
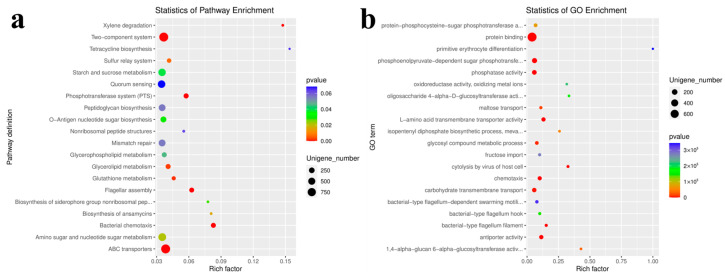
Pathways that are highly enriched in the cecum of various local chicken breeds. (**a**) GO entries; (**b**) KEGG pathways.

## Data Availability

Data are available in the manuscript.

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
