# Peer review of "Metagenomic Sequencing Revealed Differences in the Composition of Cecal Microbes in Different Breeds of Chickens"

_animals, 2023, doi:10.3390/ani14010028_

Round 1

Reviewer 1 Report

Comments and Suggestions for Authors

The study investigated that metagenomic sequencing revealed differences in the composition of cecal microbes in different breeds of chickens. The trial is interesting; however, the following points need to make clear.

1.         Animal: How the five different breed chickens were obtained from the institute, egg, pullet, adult hens? How old were they when the author received the chickens?

2.         It is not clear “all chickens were raised under the same conditions.” Did they rear under one breeding room or individual cage per breed? These points need to be described more carefully.

3.         Regarding the above comments, if the birds were reared under the same condition for only a few days after receiving the birds, the intestinal microbiota may be due to the breeding conditions of the previous feeding place (gene bank). This point is also related to the ages of birds. It takes 4-5 weeks at least to make the bacterial in the intestine stable post-hatch. Moreover, it is unclear how long it takes for the intestinal microbiota to change in adult birds in the new breeding environment. In this point, it is important to describe how long the birds were reared under the same conditions.

4.         It is hard to see all figure, so they need to be show bigger.

5.         The study just compared the cecal microbiota of five different breeds, while it is unclear that the biological characteristics such as metabolism, physiology, and immunity, behavior, and meat- or egg-type of the birds. It is required that the bacterial compositions need to refer the above characteristics for more development of the investigation.

Author Response

Comments 1.         Animal: How the five different breed chickens were obtained from the institute, egg, pullet, adult hens? How old were they when the author received the chickens?

Response 1. In this study, we selected 300-days adult hens from poultry institute as experiment materiel.

Comments 2.         It is not clear “all chickens were raised under the same conditions.” Did they rear under one breeding room or individual cage per breed? These points need to be described more carefully.

Response 2. All chickens in this experiment were hatched at same period, then we injected vaccine for them at the same day. After grew up, all hens were raised in individual cages with enough layer feed and water.

The component of feed for all chickens is the same. All cages were placed in one breeding room.

Comments 3.         Regarding the above comments, if the birds were reared under the same condition for only a few days after receiving the birds, the intestinal microbiota may be due to the breeding conditions of the previous feeding place (gene bank). This point is also related to the ages of birds. It takes 4-5 weeks at least to make the bacterial in the intestine stable post-hatch. Moreover, it is unclear how long it takes for the intestinal microbiota to change in adult birds in the new breeding environment. In this point, it is important to describe how long the birds were reared under the same conditions.

Response 3. Our gene bank is a part of poultry institute, with the obligation to make sure that all good traits of local breeds were preserved and exploited. The gene bank of poultry institute keeps variety of Chinese local chicken breeds. We hatched eggs and fed them into 300-days adult chickens. All hens were butchered at the lab of poultry institute. Then we gathered cecal content and stored them in ultra-low temperature freezer. In the whole process, all birds were reared under the same conditions.

Comments 4.         It is hard to see all figure, so they need to be show bigger.

Response 4. Yes, we have changed figures.

Comments 5.         The study just compared the cecal microbiota of five different breeds, while it is unclear that the biological characteristics such as metabolism, physiology, and immunity, behavior, and meat- or egg-type of the birds. It is required that the bacterial compositions need to refer the above characteristics for more development of the investigation.

Response 5. In this study, we did not measure the biological characteristics of chickens, because the breeds we selected are famous local chicken pedigrees with stable phenotype. Past research has revealed the biological characteristics of each breed and there is no need for us to repeat it again.

Reviewer 2 Report

Comments and Suggestions for Authors

The manuscript of Yan and colleges focuses on the differences in cecal microbiota between different Chinese chicken breeds. The differences were determined by metagenomic sequencing.

The paper is interesting and relatively well written. There are few issues that need to be addressed:

1)      There is enough information about poultry microbiome published and there is no need to cite mammalian data (lines 49-59).

2)      Why Qiime1 was used. Qiime1 is outdated.

3)      Was there any statistical analysis performed to determine the differences in taxonomic composition between breeds (Figure 2). Description of results for Figure 2 should not include any comparison, since there is no statistical analysis provided.

4)      Figure 4: please change the graph type to allow to decipher statistical differences between breeds.

5)      All Figures: it would be great if the characters for breeds were replaced by breed name or breed name abbreviation.

Comments on the Quality of English Language

N/A

Author Response

Comments 1)      There is enough information about poultry microbiome published and there is no need to cite mammalian data (lines 49-59).

Response 1. In the writing process, we gathered many examples of different species, and we think they are instructive for this article. Finally, we added all of them in background and reference list.

Comments 2)      Why Qiime1 was used. Qiime1 is outdated.

Response 2. Our colleague used to this software. Thank you for your reminding. We used Qiime2 analyzing it again, and the new result shows little difference.

Comments 3)      Was there any statistical analysis performed to determine the differences in taxonomic composition between breeds (Figure 2). Description of results for Figure 2 should not include any comparison, since there is no statistical analysis provided.

Response 3. There is no statistical analysis in this part, we have removed comparison.

Comments 4)      Figure 4: please change the graph type to allow to decipher statistical differences between breeds.

Response 4. Yes, we are going to show our results in a new way.

Comments 5)      All Figures: it would be great if the characters for breeds were replaced by breed name or breed name abbreviation.

Response 5. Agree, we have changed the abbreviation.

Reviewer 3 Report

Comments and Suggestions for Authors

Intestinal microorganisms are closely associated with health condition, and its disturbing is often affect on whole organism. Gut are colonized by diverse microbiota, including bacteria and fungi, that can be pathogenic under particular circumstances. However, differences in the composition of cecal microbes have already been evaluated.

Traditionally, microorganisms have been identified by culture-dependent methods; however, many species are fastidious and underrepresented in cultures from mixed microbial communities, whereas others cannot be cultivated under known conditions. Therefore, culture-independent molecular techniques have been used for the identification of microbial species within ecosystems.

Composition of cecal microbiota is popular topis of study:

•             Metagenomic analysis reveals linkages between cecal microbiota and

feed efficiency in Xiayan chickens PMID: 33248623 PMCID: PMC7705039 DOI:

10.1016/j.psj.2020.09.076

•             Extensive microbial and functional diversity within the chicken cecal

microbiome PLoS ONE, 9 (3) (2014), Article e91941

•             Gilroy R, Ravi A, Getino M, Pursley I, Horton DL, Alikhan NF, Baker D,

Gharbi K, Hall N, Watson M, Adriaenssens EM, Foster-Nyarko E, Jarju S, Secka A, Antonio M, Oren A, Chaudhuri RR, La Ragione R, Hildebrand F, Pallen MJ. Extensive microbial diversity within the chicken gut microbiome revealed by metagenomics and culture. PeerJ. doi:

10.7717/peerj.10941. PMID: 33868800; PMCID: PMC8035907. Silkie hens

•             Recovering High-Quality Host Genomes from Gut Metagenomic Data through

Genotype Imputation https://doi.org/10.1002/ggn2.202100065

•             Xu Y, Huang Y, Guo L, Zhang S, Wu R, Fang X, Xu H, Nie Q. Metagenomic

analysis reveals the microbiome and antibiotic resistance genes in indigenous Chinese yellow-feathered chickens. Front Microbiol. 2022 Sep 7;13:930289. doi: 10.3389/fmicb.2022.930289. PMID: 36160245; PMCID:

PMC9490229.

•             Lei Cui, Xiaolong Zhang, Ranran Cheng, Abdur Rahman Ansari,

Abdelmotaleb A. Elokil, Yafang Hu, Yan Chen, Abdallah A. Nafady, Huazhen Liu,

•             Sex differences in growth performance are related to cecal microbiota

in chicken, Microbial Pathogenesis,

https://doi.org/10.1016/j.micpath.2020.104710.

•             Yang J, Tong C, Xiao D, Xie L, Zhao R, Huo Z, Tang Z, Hao J, Zeng Z,

Xiong W. Metagenomic Insights into Chicken Gut Antibiotic Resistomes and Microbiomes. Microbiol Spectr. 2022 Apr 27;10(2):e0190721. doi:

10.1128/spectrum.01907-21. Epub 2022 Mar 1. PMID: 35230155; PMCID:

PMC9045286.

•             Yang J, Tong C, Xiao D, Xie L, Zhao R, Huo Z, Tang Z, Hao J, Zeng Z,

Xiong W. Metagenomic Insights into Chicken Gut Antibiotic Resistomes and Microbiomes. Microbiol Spectr. 2022 Apr 27;10(2):e0190721. doi:

10.1128/spectrum.01907-21. Epub 2022 Mar 1. PMID: 35230155; PMCID:

PMC9045286.

Etc.

Moreover Authors (Wei Han, Qian Xue) previously studied significant differences in gut microbiota among different chicken populations sampled from across China.

Major comments:

Abstract is difficult to understand.

Lack of Ethics Approval

Is DNA concentration and purity was evaluated?  If yes, please add this information

If Vancomycin resistance genes were identified, please add in which bacteria

Figure 2. is unreadable, same Figure 3

In cecum microbiomes lack of E. coli. Please explain why?

What about anaerobic Clostridium, fungi, parasites or viruses? Is in study population lack of these microorganism?

Line 254 – According Authors Silkies are a unique local chicken breed in China. In Europe Silkies are often breed as ornamental chickens.

Line 266 Mucispirillum schaedleri is associated with intestine inflammation in  mouse models and associated with inflammation state in intestinum. M. schaedleri is causally linked to the development of Crohn's disease-like colitis in immunodeficient mice. Is the same is in chicken?

Line 317 – Authors concluded that  explore correlation between the host and the intestinal microbiota. Lack of this data or explore I manuscript. Authors described only presence of bacteria.

Minor comments:

Line 31 -  Exiguobacterium

Line 51,52 . El-Sayed M et al. – lack of reference Line  59 - Fusobacterium Line 60 - Compared to mammals, poultry have shorter intestines. Please add more details.

Line 66,67 - Methanobrevibacter and Mucispirillum schaedler Line 143 - Firmicutes, Proteobacteria, and Bacteroidetes Line 217 - Lactobacillus aviarius and Mucispirillum schaedleri Line 220 -  Lactobacillus salivarius Line 221 - Firmicutes_bacterium_AM29_6AC, Turicibacter, Aeromonas, and Roseburia Line 253 – Clostridium Line 258, 259 - Escherichia coli, Streptococcus, Vibrio, and Clostridium Line  272 – Salmonella Line  279 -  Lactobacillus Line  286 - Lactobacillus salivarius

Line   287 – Roseburia

Line   302, 304, 305 - Streptococcus

All references should be change, especially all Authors should be added (instead of et al), Volume should be italic, and year bold

Round 2

Reviewer 1 Report

Comments and Suggestions for Authors

The Authors adequately responded to the comments.

Author Response

Thank you for your Comments!